# Kinematic Calibration of Parallel Robots Based on L-Infinity Parameter Estimation

**Dayong Yu** 

School of Mechanical Engineering, University of Shanghai for Science and Technology, Shanghai 200093, China; yudayong@usst.edu.cn

**Abstract:** Pose accuracy is one of the most important problems in the application of parallel robots. In order to adhere to strict pose error bounds, a new kinematic calibration method is proposed, which includes a new pose error model with 60 error parameters and a different kinematic parameter error identification algorithm based on L-infinity parameter estimation. Parameter errors are identified by using linear programming to minimize the maximum difference between predictions and workspace measurements. Simulation results show that the proposed kinematic calibration has better kinematic parameter error estimation and fewer pose errors when measurement noise is less than kinematic parameter errors. Experimental results show that maximum position and orientation errors, respectively, based on the proposed method are decreased by 86.48% and 87.85% of the original values and by 14.32% and 18.23% of those based on the conventional least squares method. The feasibility and validity of the proposed kinematic calibration are verified by improved pose accuracy of the parallel robot.

**Keywords:** kinematic calibration; parallel robot; parameter estimation; error model; pose accuracy

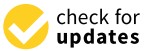



## 1. Introduction

Parallel robots have higher carrying capacity, greater structural rigidity and better dynamic response than traditional serial robots. Parallel robots have been widely applied in motion simulators, machine tools and medical devices. The pose accuracy of parallel robots is required to be higher and higher in the fields of motion simulation [1,2], mechanical manufacturing [3,4] and surgery [5,6]. The pose accuracy of parallel robots is one of the most important performance measures in the above fields.

The pose accuracy of parallel robots is affected by geometric errors [7–11] and nongeometric errors [12–16]. Geometric errors are mainly caused by manufacturing tolerances and assembly errors. Nongeometric errors might result from clearance, friction, deformation, and so on. Previous studies have shown that geometric errors are the dominant factor leading to pose inaccuracy of parallel robots. It is important for parallel robots to promote pose accuracy in practical application. Pose accuracy improvement in parallel robots is divided into accuracy analysis, synthesis and kinematic calibration.

Accuracy analysis evaluates whether the pose performance of parallel robots meets the design specifications and identifies sensitive factors affecting pose accuracy based on geometric error. An analytical method for the forward and inverse error bound analyses of a Stewart platform was developed by Kim et al. [17]. The relationship between the Stewart platform pose errors and the joint space errors is characterized by the kinematic error model. The forward and inverse error bound are obtained by solving two eigenvalue problems. Comprehensive accuracy modeling and analysis of a new type of lock-or-release mechanism was proposed by Ding et al. [18]. Two accuracy models were established and verified by Monte Carlo simulation and an experiment designed to influence factor sensitivities, and results show that the manufacturing tolerances of a lead screw are the most significant influence factor. Accuracy analysis of a parallel positioning mechanism with actuation

redundancy was investigated by Ding et al. [19]. The effects of input uncertainty, components stiffness and redundant limbs were addressed; mean value and standard deviation of the pose errors were computed by optimal Latin hypercube sampling algorithm.

Accuracy synthesis optimally allocates component tolerances of parallel robots under different assembly indices according to the design specification. Accuracy synthesis of a multi-level hybrid positioning mechanism was studied by Tang et al. [20]. Three types of error influence factors are considered in the error model, and the error boundary of the multi-level hybrid positioning mechanism is obtained by using the vector set theory and a linear algebra method. Accuracy synthesis was performed based on a nonlinear optimization algorithm. A comprehensive methodology for implementing the required pose accuracy of a 4-DOF parallel robot was presented by Huang et al. [21]. In this work, all possible geometric errors were separated as either identifiable or unidentifiable geometric errors. The unidentifiable geometric errors were restrained by tolerance design and assembly. Pose accuracy in the whole workspace was achieved by a linear and real-time error compensator. A systematic tolerance design method of parallel link robots was proposed by Takematsu et al. [22]. The standard deviations of the kinematic motions of the end effector were represented by the tolerance values of all joints and links. A suitable set the tolerance values for all joints and links was determined using an optimization algorithm.

Kinematic calibration achieves an inverse kinematic model that more closely matches the actual system in all possible configurations. In general, kinematic calibration can be divided into four steps: error modeling, pose measurement, parameter identification and error compensation. Kinematic calibration can be classified into two categories: external calibration and self-calibration. Kinematic calibration of a Stewart platform was presented by Zhuang et al. [23]. Kinematic error parameters of the Stewart platform were identified using the Gauss–Newton algorithm, and the kinematic error parameters of each leg were solved independently. However, precise pose measurement needs be performed in this approach. Daney [24] established a complete kinematic model of the Gough platform and a unified kinematic parameter identification scheme, and presented an original kinematic calibration method based on the above principle. The accuracy of the Hexapode 300 was experimentally improved by 99% using the original kinematic calibration. A novel identifiable parameter separation method for kinematic calibration of a 6-DOF parallel manipulator was proposed by Hu et al. [25]. The method can reduce the number of kinematic error parameters in the identification model and improve the convergence of the parameter identification algorithm by simple and direct measuring. A systematic kinematic calibration method of a 6-DOF hybrid polishing robot was presented by Huang et al. [26]. Ill-conditioning of the identification Jacobian was dealt with by establishing a linear regression model and implementing kinematic error parameter estimation and pose error compensation using a linear least squares algorithm and Liu estimator. A new error model based on a dimensionless error mapping matrix for kinematic calibration of a 5-axis parallel machining robot was proposed by Luo et al. [27]. Kinematic error parameters are unified into the same unit in the error model and are identified by an iterative least squares procedure based on full pose measurement with a laser tracker. A comprehensive error model for kinematic calibration of a non-fully symmetric parallel Delta robot was presented by Shen et al. [28]. Variations of the parallel Delta robot components and geometric parameters were considered in the error identification model, and the variations were identified by a least squares algorithm.

Kinematic error parameter identification in kinematic calibration can be treated as the best approximation of measurement data. Large amounts of research have been reported on kinematic error parameter identification based on various identification algorithms [29–32]. A novel geometric calibration of industrial robots was presented by Wu et al. [33]. The design of experiments was proposed and added to the conventional kinematic calibration procedure. The additional step is performed before pose measurement in order to obtain a set of optimal measurement poses that ensure the best robot positioning accuracy after kinematic calibration. A dedicated geometric parameter identification algorithm was de-

scribed, and the identification procedure was divided into two steps. These two steps were repeated iteratively to achieve the desired geometric parameter identification accuracy. A robust kinematic calibration of serial robots based on separable nonlinear least squares was proposed by Mao et al. [34]. The optimal geometric parameter identification problem was converted into a separable nonlinear least squares problem by using the distinctive characteristic of the MDH model. Kinematic calibration of industrial robots based on distance measurement information was presented by Gao et al. [35]. A novel extended Kalman filter and regularized particle filter hybrid identification algorithm was adopted to identify the kinematic parameters of the linearized error model. The algorithm solved the problem with traditional optimization algorithms of being easily affected by measurement noise in high-dimension identification. However, there is little in the literature on reduction of the impact of measurement noise by selecting optimal measurement poses in kinematic calibration. In order to compare the different pose measurement schemes, several observability measures were presented in [36–40] and were used to choose an optimal pose measurement scheme. These measures are not directly related to the pose accuracy of kinematic calibration. A new industry-oriented performance measure is presented in [33] with the intent of ensuring the best robot positioning accuracy after geometric error compensation.

The least squares algorithm has been universally used to identifying kinematic error parameters of parallel robots from pose measurements. Kinematic error can be identified, analyzed and corrected to minimize the sum of squares of the difference between measured errors and computed errors. Although this algorithm is mathematically convenient and can achieve better average pose accuracy in a parallel robot workspace, it may result in pose accuracy not being evenly distributed in the workspace and may even lead to large pose errors outside of the subset.

In order to improve the uneven distribution of pose accuracy and to reduce large pose error, a new kinematic calibration method for parallel robots is presented based on L-infinity parameter estimation and applied to the spacecraft docking motion simulation system. The paper is organized as follows: An inverse kinematic model of the parallel robot is described, and a forward kinematic solution is presented in Section 2. A pose error model for kinematic calibration is established in Section 3. A new kinematic parameter error identification algorithm based on L-infinity parameter estimation is proposed in Section 4. Simulations and experiments are performed, and the results are shown in Section 5. Finally, some conclusions are given in Section 6.

## 2. Kinematic Model

A parallel robot model is composed of a moving platform, a base, and six identical hydraulic cylinders with variable lengths, as shown in Figure 1. The moving platform's position relative to the base can be controlled by varying the length of the six hydraulic cylinders. The parallel robot has six DOF. The base coordinate system $O_B$-*xyz* is located in the center of the base. The mobile coordinate system $O_P$-*xyz* is attached to the center of the moving platform. All vectors and matrices will be denoted in bold letters. The two coordinate systems $O_B$-*xyz* and $O_P$-*xyz* can be related to each other through a vector $\mathbf{q} = \begin{bmatrix} x & y & z & \phi & \theta & \psi \end{bmatrix}^T$ that describes the pose of the moving platform by its position (longitudinal ($x$), lateral ($y$) and vertical ($z$) displacements) and its orientation (Roll ($\phi$), Pitch ($\theta$) and Yaw ($\Psi$) angles). Thus, the position of the moving platform can be expressed by a position vector $\mathbf{t}$ as

$$\mathbf{t} = \begin{bmatrix} x & y & z \end{bmatrix}^T \tag{1}$$

and the orientation of the moving platform can be expressed by a rotation matrix $\mathbf{R}$ as

$$\mathbf{R} = \begin{bmatrix} \cos\psi\cos\theta & \cos\psi\sin\theta\sin\phi - \sin\psi\cos\phi & \cos\psi\sin\theta\cos\phi + \sin\psi\sin\phi \\ \sin\psi\cos\theta & \sin\psi\sin\theta\sin\phi + \cos\psi\cos\phi & \sin\psi\sin\theta\cos\phi - \cos\psi\sin\phi \\ -\sin\theta & \cos\theta\sin\phi & \cos\theta\cos\phi \end{bmatrix} \tag{2}$$

where $\phi$, $\theta$, and $\psi$ are three Roll, Pitch and Yaw (RPY) angles chosen with respect to the $x$-axes, the $y$-axes and the $z$-axes, respectively, of the base coordinate system $O_B$-$xyz$.

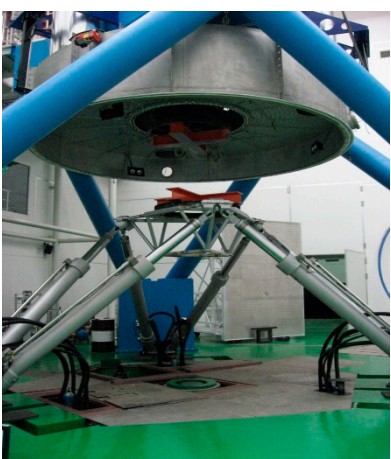

**Figure 1.** The parallel robot.

### 2.1. Inverse Kinematics

Referring to Figures 2 and 3, $\mathbf{u}_i$ is the unit vector along the $i$th hydraulic cylinder direction, and $l_i$ is the length of the $i$th hydraulic cylinder; $\mathbf{a}_i$ is the position vector from $O_P$ to $A_i$ and is represented in the mobile coordinate system $O_P$-xyz, and $\mathbf{b}_i$ is the position vector from $O_B$ to $B_i$ and is represented in the base coordinate system $O_B$-xyz. A vector chain equation can be expressed as

$$l_i\mathbf{u}_i = \mathbf{R}\mathbf{a}_i + \mathbf{t} - \mathbf{b}_i \; i = 1, 2, \ldots, 6 \tag{3}$$

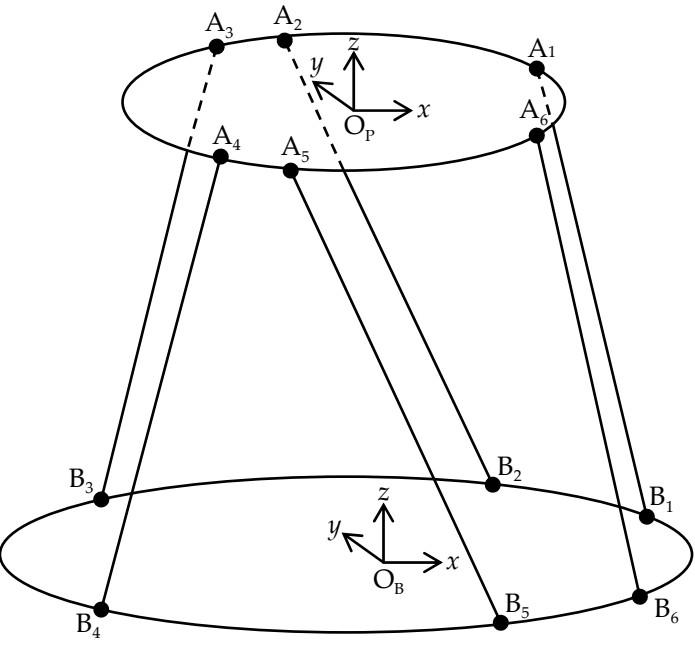

**Figure 2.** Coordinate system of the parallel robot.

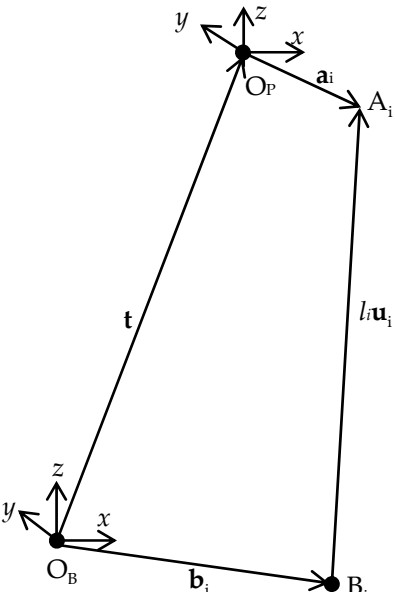

**Figure 3.** Vector chain for hydraulic cylinder *i*.

The vector chain equation is derived for the perfect (no errors) parallel robot. The length of the *i*th hydraulic cylinder can be computed from

$$l_i = f_i(\mathbf{R}, \mathbf{t}) = \sqrt{(\mathbf{R}\mathbf{a}_i + \mathbf{t} - \mathbf{b}_i)^T(\mathbf{R}\mathbf{a}_i + \mathbf{t} - \mathbf{b}_i)} \ i = 1, 2, \ldots, 6 \tag{4}$$

and the measured length of the *i*th hydraulic cylinder can be obtained by

$$s_i = l_i - l_{0,i} \ i = 1, 2, \ldots, 6 \tag{5}$$

where $l_{0,i}$ is the initial length of the *i*th hydraulic cylinder.

*2.2. Forward Kinematics*

The forward kinematics of the parallel robot compute the moving platform pose when the measured hydraulic cylinder lengths are given and the kinematic parameters are known. Although the inverse kinematics for the parallel robot can be expressed in a closed form, forward kinematics offer no analytical solution. Mapping the pose using the hydraulic cylinder lengths is complicated to solve (Equation (4)). Numerical methods are often employed to solve the forward kinematics for parallel robots. The following method for the forward kinematics of a parallel robot is based on the Newton–Raphson algorithm.

For solving the forward kinematics of a parallel robot, a vector function is defined to describe the difference between the estimated hydraulic cylinder length $s_{ei}$ and the measured hydraulic cylinder length $s_{ai}$.

$$\mathbf{f} = \begin{bmatrix} f_1 \\ \vdots \\ f_6 \end{bmatrix} = \begin{bmatrix} s_{e1}^2 - s_{a1}^2 \\ \vdots \\ s_{e6}^2 - s_{a6}^2 \end{bmatrix} \tag{6}$$

The Newton–Raphson algorithm can be stated as:

(1)  Measure $s_{ai}$, and select an initial guess for the pose, $\mathbf{q}$.
(2)  Compute $s_{ei}$ based on $\mathbf{q}$.
(3)  Form $\mathbf{f}$.
(4)  If $\mathbf{q}^T\mathbf{q} < \text{tolerance}_1$, exit with $\mathbf{q}$ as the solution.
(5)  Compute the partial derivative matrix $\mathbf{J} = \frac{\partial \mathbf{f}}{\partial \mathbf{q}}$ such that $\mathbf{J}_{i,j} = \frac{\partial f_i}{\partial q_j}$.

(6) Obtain update $\delta\mathbf{q}$ by solving $\mathbf{J}\delta\mathbf{q} = -\mathbf{f}$.

(7) If $\delta\mathbf{q}^T\delta\mathbf{q} < \text{tolerance}_2$, exit with $\mathbf{q}$ as the solution.

(8) Update $\mathbf{q}$ by $\mathbf{q} = \mathbf{q} + \delta\mathbf{q}$ and go to step (2).

The accuracy and rate convergence for the Newton–Raphson algorithm depend on several factors. The algorithm rapidly converges if the initial guess is in the neighborhood of the solution, and the algorithm is fairly robust with the choice of the initial guess. If the second order terms are large, this first order approach will not be accurate, and the algorithm will converge very slowly. The existence of the Jacbian inverse is required in step (6), and thus the moving platform may not be near a singularity configuration. If convergence problems arise, or if speed is of paramount importance, the forward kinematics may require a different algorithm, such as those presented in [41–44].

## 3. Error Model

The kinematic parameters of the parallel robot might be different from those in the design specification due to imprecision in manufacturing and assembly of the joints and the initial length of the hydraulic cylinders. The difference will lead to pose error of the moving platform. An error model relating the kinematic parameter errors to the pose errors is derived in this section.

A vector differential error model is obtained by performing the following differentiation for Equation (3) as

$$\delta l_i \mathbf{u}_i + l_i \delta\mathbf{u}_i = \delta\mathbf{R}\mathbf{a}_i + \mathbf{R}\delta\mathbf{a}_i + \delta\mathbf{t} - \delta\mathbf{b}_i \quad i = 1, 2, \ldots, 6 \tag{7}$$

where $\delta l_i$ is the length error of $l_i$, $\delta\mathbf{u}_i$ is the deviation vector of $\mathbf{u}_i$, $\delta\mathbf{R}$ is the deviation matrix of $\mathbf{R}$, $\delta\mathbf{a}_i$ is the position error vector of $\mathbf{a}_i$, $\delta\mathbf{t}$ is a deviation vector of $\mathbf{t}$, and $\delta\mathbf{b}_i$ is the position error vector of $\mathbf{b}_i$.

The deviation vector $\delta\mathbf{u}_i$ can be expressed as

$$\delta\mathbf{u}_i = \Delta_{\mathbf{u}_i}\mathbf{u}_i = \begin{bmatrix} 0 & -\delta u_{iz} & \delta u_{iy} \\ \delta u_{iz} & 0 & -\delta u_{ix} \\ -\delta u_{iy} & \delta u_{ix} & 0 \end{bmatrix} \begin{bmatrix} u_{ix} \\ u_{iy} \\ u_{iz} \end{bmatrix} \tag{8}$$

where $\Delta_{\mathbf{u}_i}$ is a skew symmetric matrix of $\delta\mathbf{u}_i$.

Let $\delta\boldsymbol{\omega}$ be the angular error vector of the nominal RPY angles $\phi$, $\theta$ and $\psi$, and be represented in the base coordinate system. The angular error vector $\delta\boldsymbol{\omega}$ can be expressed as

$$\delta\boldsymbol{\omega} = \begin{bmatrix} \delta\omega_x \\ \delta\omega_y \\ \delta\omega_z \end{bmatrix} = \begin{bmatrix} -\sin\psi\delta\theta + \cos\psi\cos\theta\delta\phi \\ \cos\psi\delta\theta + \sin\psi\cos\theta\delta\phi \\ -\sin\theta\delta\phi + \delta\psi \end{bmatrix} \tag{9}$$

The skew symmetric matrix of $\delta\boldsymbol{\omega}$ can be written as

$$\Delta_{\boldsymbol{\omega}} = \begin{bmatrix} 0 & -\delta\omega_z & \delta\omega_y \\ \delta\omega_z & 0 & -\delta\omega_x \\ -\delta\omega_y & \delta\omega_x & 0 \end{bmatrix} \tag{10}$$

The deviation matrix $\delta\mathbf{R}$ can be given by

$$\delta\mathbf{R} = \Delta_{\boldsymbol{\omega}}\mathbf{R} = \begin{bmatrix} 0 & \sin\theta\delta\phi - \delta\psi & \cos\psi\delta\theta + \sin\psi\cos\theta\delta\phi \\ -\sin\theta\delta\phi + \delta\psi & 0 & \sin\psi\delta\theta - \cos\psi\cos\theta\delta\phi \\ -\cos\psi\delta\theta - \sin\psi\cos\theta\delta\phi & -\sin\psi\delta\theta + \cos\psi\cos\theta\delta\phi & 0 \end{bmatrix}\mathbf{R} \tag{11}$$

Substituting Equations (8) and (11) into Equation (7) yields

$$\delta l_i \mathbf{u}_i + l_i \Delta_{\mathbf{u}_i}\mathbf{u}_i = \Delta_{\boldsymbol{\omega}}\mathbf{R}\mathbf{a}_i + \mathbf{R}\delta\mathbf{a}_i + \delta\mathbf{t} - \delta\mathbf{b}_i \quad i = 1, 2, \ldots, 6 \tag{12}$$

Let $\mathbf{a}'_i = \mathbf{R}\mathbf{a}_i$, Equation (12) can be rewritten as

$$\delta l_i \mathbf{u}_i + l_i \Delta_{\mathbf{u}_i} \mathbf{u}_i = \Delta_{\boldsymbol{\omega}} \mathbf{a}'_i + \mathbf{R}\delta \mathbf{a}_i + \delta \mathbf{t} - \delta \mathbf{b}_i \; i = 1, 2, \ldots, 6 \tag{13}$$

Equation (13) can be expressed in matrix form as

$$\begin{bmatrix} \mathbf{I} & \Delta_{\mathbf{a}'_i}^T \end{bmatrix} \begin{bmatrix} \delta \mathbf{t} \\ \delta \boldsymbol{\omega} \end{bmatrix} = \begin{bmatrix} \mathbf{u}_i & l_i \Delta_{\mathbf{u}'_i}^T & -\mathbf{R} & \mathbf{I} \end{bmatrix} \begin{bmatrix} \delta l_i \\ \delta \mathbf{u}_i \\ \delta \mathbf{a}_i \\ \delta \mathbf{b}_i \end{bmatrix} \; i = 1, 2, \ldots, 6 \tag{14}$$

where $\mathbf{I}$ is $3 \times 3$ unit matrix, $\Delta_{\mathbf{a}'_i}$ is a skew symmetric matrix of $\mathbf{a}'_i$, and $\Delta_{\mathbf{u}'_i}$ is a skew symmetric matrix of $\mathbf{u}_i$.

Equation (14) can be rewritten as

$$\mathbf{J}_{\Omega_i} \delta \Omega = \mathbf{J}_i \delta \mathbf{p}_i \; i = 1, 2, \ldots, 6 \tag{15}$$

where

$$\delta \Omega = \begin{bmatrix} \delta \mathbf{t}^T & \delta \boldsymbol{\omega}^T \end{bmatrix}^T = \begin{bmatrix} \delta x & \delta y & \delta z & \delta \omega_x & \delta \omega_y & \delta \omega_z \end{bmatrix}^T \tag{16}$$

represents the pose error of the parallel robot, and the following matrices, $\mathbf{J}_{\Omega_i}$ and $\mathbf{J}_i$ are the inverse and forward error mapping components defined as

$$\mathbf{J}_{\Omega_i} = \begin{bmatrix} \mathbf{I} & \Delta_{\mathbf{a}'_i}^T \end{bmatrix} \tag{17}$$

$$\mathbf{J}_i = \begin{bmatrix} \mathbf{u}_i & l_i \Delta_{\mathbf{u}'_i}^T & -\mathbf{R} & \mathbf{I} \end{bmatrix} \tag{18}$$

and

$$\delta \mathbf{p}_i = \begin{bmatrix} \delta l_i & \delta u_{ix} & \delta u_{iy} & \delta u_{iz} & \delta a_{ix} & \delta a_{iy} & \delta a_{iz} & \delta b_{ix} & \delta b_{iy} & \delta b_{iz} \end{bmatrix}^T \tag{19}$$

represents the kinematic parameter errors in the individual vector chain.

Considering all six vector chains, Equation (14) can be expressed in the following matrix form

$$\begin{bmatrix} \mathbf{I} & \Delta_{\mathbf{a}'_1}^T \\ \mathbf{I} & \Delta_{\mathbf{a}'_2}^T \\ \mathbf{I} & \Delta_{\mathbf{a}'_3}^T \\ \mathbf{I} & \Delta_{\mathbf{a}'_4}^T \\ \mathbf{I} & \Delta_{\mathbf{a}'_5}^T \\ \mathbf{I} & \Delta_{\mathbf{a}'_6}^T \end{bmatrix} \begin{bmatrix} \delta \mathbf{t} \\ \delta \boldsymbol{\omega} \end{bmatrix} = \begin{bmatrix} \mathbf{J}_1 & & & & & \\ & \mathbf{J}_2 & & & & \\ & & \mathbf{J}_3 & & & \\ & & & \mathbf{J}_4 & & \\ & & & & \mathbf{J}_5 & \\ & & & & & \mathbf{J}_6 \end{bmatrix} \begin{bmatrix} \delta \mathbf{p}_1 \\ \delta \mathbf{p}_2 \\ \delta \mathbf{p}_3 \\ \delta \mathbf{p}_4 \\ \delta \mathbf{p}_5 \\ \delta \mathbf{p}_6 \end{bmatrix} \tag{20}$$

Equation (20) above can be rewritten as

$$\mathbf{J}_\Omega \delta \Omega = \mathbf{J}_\mathbf{p} \delta \mathbf{p} \tag{21}$$

where $\mathbf{J}_\Omega$ represents the inverse error mapping matrix of the parallel robot, $\mathbf{J}_\mathbf{p}$ represents the forward error mapping matrix, and $\delta \mathbf{p}$ represents the kinematic parameter errors for all the vector chains. The vector $\delta \mathbf{p}$ contains 60 linearly independent error parameters, and the $j$th element of the vector can be denoted as $\delta p_j$.

The pose error of the parallel robot can be computed by

$$\delta \Omega = \mathbf{J} \delta \mathbf{p} \tag{22}$$

where

$$\mathbf{J} = \left( \mathbf{J}_\Omega^T \mathbf{J}_\Omega \right)^{-1} \mathbf{J}_\Omega^T \mathbf{J}_\mathbf{p} \tag{23}$$

is defined as the error Jacobian matrix for the parallel robot, and its condition number will be used to choose the optimal pose measurement configurations.

The relationship between the pose errors of the parallel robot and the kinematic parameter errors is described by Equation (22). It is a linear equation in terms of the unknown kinematic parameter errors, which can be identified based on L-infinity parameter estimation once the pose errors of the parallel robot are measured.

## 4. Calibration Method

The least squares fit is universally used to identify kinematic parameter errors from measurement data in kinematic calibration. Kinematic error can be identified, analyzed and corrected to minimize the sum of squares of the difference between measured errors and computed errors. Thus, for a parallel robot using a control model compensated with kinematic parameter errors and measuring a number of poses in its workspace, nothing can be said of its accuracy at any one pose. If the sample of poses measured represents an unbiased sample of the workspace, the mean squares errors of the parallel robot at these poses is minimized. That is, the least squares fit does not minimize or bound the pose error between the measured pose errors and the computed pose errors based on the error model at a single pose.

The parallel robot is used with the spacecraft docking motion simulation system, so its pose accuracy will be evaluated not on the basis of average error of all poses on a simulated trajectory, but based on the error of each pose of a simulated trajectory meeting a given accuracy specification. In order to achieve the given accuracy requirement at any one pose in the whole workspace, a different kinematic parameter error identification algorithm based on L-infinity parameter estimation is selected. It identifies kinematic parameter errors of the parallel robot by minimizing the maximum difference between measured pose errors and computed pose errors based on an error model and can bound large pose errors and equalize pose errors across the workspace. Unknown kinematic parameter errors of the error model (Equation (22)) can be identified by the following formulation based on L-infinity parameter estimation:

$$\min \max |\delta \mathbf{\Omega}_i| \tag{24}$$

where $\delta \mathbf{\Omega}_i$ is computed by

$$\delta \mathbf{\Omega}_i = \delta \mathbf{\Omega}_i^m - \delta \mathbf{\Omega}_i^c \ i = 1, 2, \ldots, n \tag{25}$$

$\delta \mathbf{\Omega}_i^m$ is the measured pose error, $\delta \mathbf{\Omega}_i^c$ is the computed pose error based on Equation (22), and $n$ represents the number of measurement poses in the workspace of the parallel robot.

Equation (22) is rewritten in terms of the kinematic parameter errors at the $i$th measurement pose as

$$\delta \Omega_{ki}^c = \sum_{j=1}^{60} J_{ki}^j \delta p_j \ k = 1, 2, \ldots, 6 \tag{26}$$

The total number of identification equations will be six times the total number of measurement poses. The index assigned to the identification equations will be $w$, and it can take values between 1 and $6n$. Substituting Equations (25) and (26) into Equation (24), the following is obtained:

$$\min \max \left| \delta \Omega_w^m - \sum_{j=1}^{60} J_w^j \delta p_j \right| \ w = 1, 2, \ldots, 6n \tag{27}$$

Equation (27) is subject to no restrictions. The L-infinity parameter identification problem can be converted to a linear programming problem with the introduction of the variable $z$, thus the following is obtained:

$$\min z = \sum_{j=1}^{60} O_j \delta p_j + y \tag{28}$$

subject to

$$y + \sum_{j=1}^{60} J_w^j \delta p_j \geq \delta\Omega_w^m \; w = 1, 2, \ldots, 6n \tag{29}$$

$$y - \sum_{j=1}^{60} J_w^j \delta p_j \geq -\delta\Omega_w^m \; w = 1, 2, \ldots, 6n \tag{30}$$

$$y = \max \left| \delta\Omega_w^m - \sum_{j=1}^{60} J_w^j \delta p_j \right| \; w = 1, 2, \ldots, 6n \tag{31}$$

The variable $y$ represents the absolute value of the maximum discrepancy between the measured pose errors and the computed pose errors based on the error model in the above linear program, and **O** represents a $1 \times 60$ zero vector. The unknown kinematic parameter errors can be identified by minimizing the variable $z$. The above linear programming problem can be solved by using the simplex method [45]. For identification of 60 kinematic parameter errors, it is expected to measure the poses that are located near or at the boundaries of the workspace, which can provide sufficient pose error vectors to expand the parameter space of the error model (Equation (22)). At least 30 measurement poses are required in kinematic calibration of the parallel robot.

## 5. Simulations and Experiments

### 5.1. Model Verification

In order to verify the pose error model derived in Section 3, a numerical simulation scheme is designed and performed by computer programs. The nominal kinematic parameters and the assumed kinematic parameter errors are listed in Tables 1 and 2, respectively. The procedure can be described as follows:

1.  Select a set of desired poses evenly distributed in the workspace.
2.  Compute the measured lengths of the six hydraulic cylinders by using inverse kinematics with the nominal kinematic parameters in Table 1.
3.  Actuate the parallel robot to the selected poses in sequence with the measured lengths of the hydraulic cylinders, and compute the actual poses by using forward kinematics with the actual kinematic parameters (the nominal kinematic parameters plus the assumed kinematic parameter errors in Table 2).
4.  Compute the actual pose errors, namely, subtract the selected poses from the actual poses.
5.  Compute the pose errors by using the pose error model with the nominal kinematic parameters, the lengths and the unit vectors of the hydraulic cylinders, and the kinematic parameter errors.
6.  Draw the contrasting curves of the position error and the orientation error for the above numerical simulation results in Figures 4 and 5.

**Table 1.** The nominal kinematic parameters.

| | $a_{ix}$ (mm) | $a_{iy}$ (mm) | $a_{iz}$ (mm) | $b_{ix}$ (mm) | $b_{iy}$ (mm) | $b_{iz}$ (mm) | $l_{ix}$ (mm) | $l_{iy}$ (mm) | $l_{iz}$ (mm) |
|---|---|---|---|---|---|---|---|---|---|
| 1 | 1394.7 | 122.0 | 0 | 2049.3 | 3038.5 | 0 | −654.6 | −2916.5 | 3091.2 |
| 2 | −591.7 | 1268.8 | 0 | 1606.8 | 3294.0 | 0 | −2198.5 | −2025.2 | 3091.2 |
| 3 | −803.0 | 1146.8 | 0 | −3656.1 | 255.5 | 0 | 2853.1 | 891.3 | 3091.2 |
| 4 | −803.0 | −1146.8 | 0 | −3656.1 | −255.5 | 0 | 2853.1 | −891.3 | 3091.2 |
| 5 | −591.7 | −1268.8 | 0 | 1606.8 | −3294.0 | 0 | −2198.5 | 2025.2 | 3091.2 |
| 6 | 1394.7 | −122.0 | 0 | 2049.3 | −3038.5 | 0 | −654.6 | 2916.5 | 3091.2 |

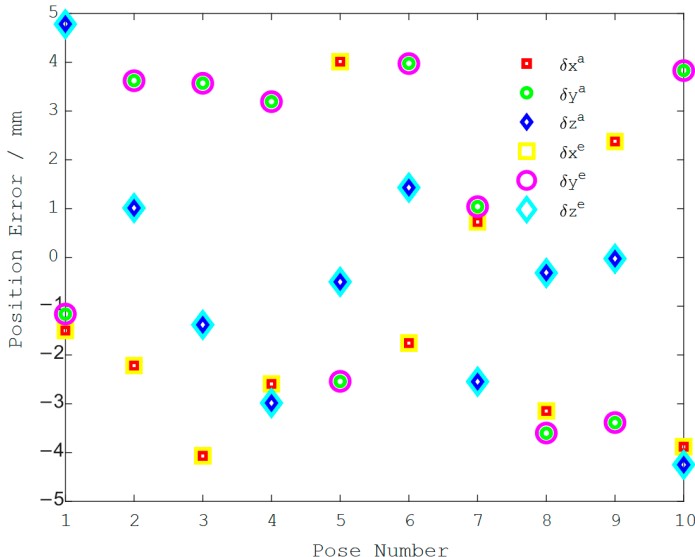

**Figure 4.** Comparison of the position error computation results.

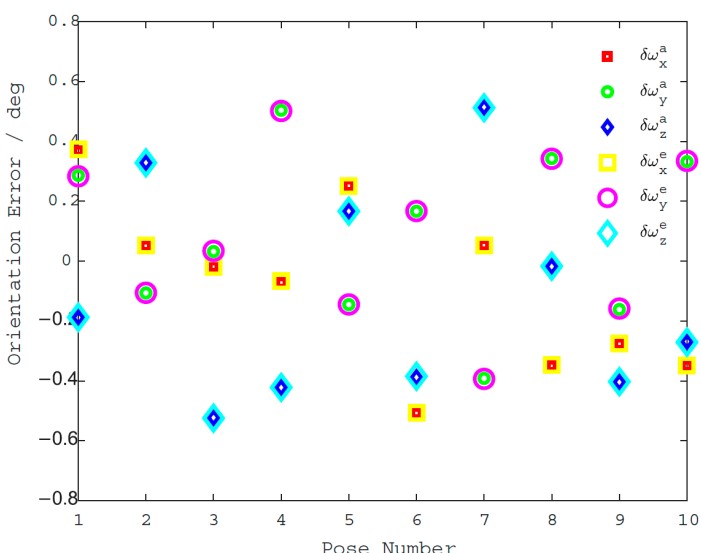

**Figure 5.** Comparison of the orientation error computation results.

**Table 2.** The assumed kinematic parameter errors.

|   | $\delta a_{ix}$ (mm) | $\delta a_{iy}$ (mm) | $\delta a_{iz}$ (mm) | $\delta b_{ix}$ (mm) | $\delta b_{iy}$ (mm) | $\delta b_{iz}$ (mm) | $\delta l_{ix}$ (mm) | $\delta l_{iy}$ (mm) | $\delta l_{iz}$ (mm) |
|---|---|---|---|---|---|---|---|---|---|
| 1 | 0.90 | −0.09 | 0.84 | −0.18 | −0.72 | −0.97 | 0.69 | 0.36 | −0.39 |
| 2 | −0.54 | −0.96 | 0.48 | 0.79 | −0.59 | 0.49 | 0.05 | −0.24 | −0.62 |
| 3 | 0.21 | 0.64 | −0.65 | −0.88 | −0.60 | −0.11 | −0.59 | 0.66 | −0.61 |
| 4 | −0.03 | −0.11 | −0.19 | −0.29 | 0.21 | 0.86 | 0.34 | 0.05 | 0.36 |
| 5 | 0.78 | 0.23 | 0.87 | 0.63 | −0.46 | −0.07 | 0.68 | 0.42 | −0.39 |
| 6 | 0.52 | 0.58 | 0.83 | −0.98 | −0.61 | −0.16 | −0.96 | −0.14 | 0.08 |

The following conclusions can be summarized from Figures 4 and 5.

1. Pose errors vary at different locations in the workspace. The pose errors are affected not only by kinematic parameter errors, but also by the pose of the parallel robot.
2. Pose errors computed by using the pose error model are basically consistent with the actual pose errors.

Therefore, the proposed pose error model is verified to be correct and to represent the kinematic parameter errors of the parallel robot.

### 5.2. Identification Simulations

Kinematic parameter error identifications were simulated with various kinematic parameter errors, measurement noise levels and pose configuration sets. The actual coordinates of the feature points of the moving platform were measured by a coordinate measuring machine, and the actual poses of the parallel robot were computed. Three kinematic parameter error sets were given, with the assumed kinematic parameter errors obtained from normal distributions with variances of 0.01 mm (set I), 0.1 mm (set II) and 1 mm (set III). These kinematic parameter error sets are shown in Tables 3–5, respectively. Gaussian noise with variances of 0.0001 mm, 0.001 mm, 0.01 mm and 0.1 mm was added to the coordinate measurements of the feature points of the moving platform to simulate measurement noise. Four different pose sets were used in the identification simulations. Pose set 1 contains 32 random poses. Pose set 2 contains 24 poses based on a full factorial exploration of the six pose variable limits. Pose set 3 contains 32 poses selected from the workspace using a coordinate exchange algorithm for optimal experimental design. Pose set 4 contains 64 poses selected using a coordinate exchange algorithm.

**Table 3.** The assumed kinematic parameter errors with variances of 0.01 mm.

|   | $\delta a_{ix}$ (mm) | $\delta a_{iy}$ (mm) | $\delta a_{iz}$ (mm) | $\delta b_{ix}$ (mm) | $\delta b_{iy}$ (mm) | $\delta b_{iz}$ (mm) | $\delta l_{ix}$ (mm) | $\delta l_{iy}$ (mm) | $\delta l_{iz}$ (mm) |
|---|---|---|---|---|---|---|---|---|---|
| 1 | 0.0538 | 0.1834 | −0.2259 | 0.0862 | 0.0319 | −0.1308 | −0.0434 | 0.0343 | 0.3578 |
| 2 | 0.2769 | −0.1350 | 0.3035 | 0.0725 | −0.0063 | 0.0715 | −0.0205 | −0.0124 | 0.1490 |
| 3 | 0.1409 | 0.1417 | 0.0671 | −0.1207 | 0.0717 | 0.1630 | 0.0489 | 0.1035 | 0.0727 |
| 4 | −0.0303 | 0.0294 | −0.0787 | 0.0888 | −0.1147 | −0.1069 | −0.0809 | −0.2944 | 0.1438 |
| 5 | 0.0325 | −0.0755 | 0.1370 | −0.1712 | −0.0102 | −0.0241 | 0.0319 | 0.0313 | −0.0865 |
| 6 | −0.0030 | −0.0165 | 0.0628 | 0.1093 | 0.1109 | −0.0864 | 0.0077 | −0.1214 | −0.1114 |

**Table 4.** The assumed kinematic parameter errors with variances of 0.1 mm.

|   | $\delta a_{ix}$ (mm) | $\delta a_{iy}$ (mm) | $\delta a_{iz}$ (mm) | $\delta b_{ix}$ (mm) | $\delta b_{iy}$ (mm) | $\delta b_{iz}$ (mm) | $\delta l_{ix}$ (mm) | $\delta l_{iy}$ (mm) | $\delta l_{iz}$ (mm) |
|---|---|---|---|---|---|---|---|---|---|
| 1 | −0.0022 | 0.4847 | −0.2434 | 0.1174 | −0.0713 | 0.3533 | −0.3444 | 0.0103 | 0.1747 |
| 2 | 0.3480 | 0.4883 | 0.0272 | −0.4717 | −0.2347 | −0.3357 | 0.7433 | −0.1947 | 0.2366 |
| 3 | −0.0608 | 0.2810 | −0.2419 | −0.4434 | −0.4498 | 0.1544 | −0.0561 | −0.0620 | 0.4488 |
| 4 | 0.0922 | 0.0626 | 0.5021 | −0.2544 | 0.2203 | 0.2641 | −0.0771 | 0.0682 | −0.3687 |
| 5 | −0.3630 | 0.0332 | 0.2284 | 0.8176 | −0.2109 | 0.0592 | −0.0261 | −0.6113 | −0.1388 |
| 6 | −0.5675 | 0.2658 | −0.2808 | 0.0317 | −0.1722 | 0.0960 | −0.1898 | 0.1549 | 0.2338 |

**Table 5.** The assumed kinematic parameter errors with variances of 1 mm.

|   | $\delta a_{ix}$ (mm) | $\delta a_{iy}$ (mm) | $\delta a_{iz}$ (mm) | $\delta b_{ix}$ (mm) | $\delta b_{iy}$ (mm) | $\delta b_{iz}$ (mm) | $\delta l_{ix}$ (mm) | $\delta l_{iy}$ (mm) | $\delta l_{iz}$ (mm) |
|---|---|---|---|---|---|---|---|---|---|
| 1 | 1.7119 | −0.1941 | −2.1384 | −0.8396 | 1.3546 | −1.0722 | 0.9610 | 0.1240 | 1.4367 |
| 2 | −1.9609 | −0.1977 | −1.2078 | 2.9080 | 0.8252 | 1.3790 | −1.0582 | −0.4686 | −0.2725 |
| 3 | 1.0984 | −0.2779 | 0.7015 | −2.0518 | −0.3538 | −0.8236 | −1.5771 | 0.5080 | 0.2820 |
| 4 | 0.0335 | −1.3337 | 1.1275 | 0.3502 | −0.2991 | 0.0229 | −0.2620 | −1.7502 | −0.2857 |
| 5 | −0.8314 | −0.9792 | −1.1564 | −0.5336 | −2.0026 | 0.9642 | 0.5201 | −0.0200 | −0.0348 |
| 6 | −0.7982 | 1.0187 | −0.1332 | −0.7145 | 1.3514 | −0.2248 | −0.5890 | −0.2938 | −0.8479 |

For each simulation, kinematic parameter errors were identified using L-infinity parameter estimation based on the LINPROG function of the MATLAB Optimization Toolbox. Kinematic parameter identification error was computed as the root mean square value of the difference between the actual kinematic parameter errors and the identified kinematic parameter errors. In order to evaluate the resulting pose accuracy improvement, pose errors were computed before and after kinematic calibration by using the pose error model given in the previous section in this paper. Position error was computed as the maximum absolute value of the error along the x, y and z axes at 100 evaluation poses randomly distributed in the workspace. Orientation error was computed as the maximum absolute value of the error around the x, y and z axes at the same poses as above. All identification simulations where the measurement noise level was less than the kinematic parameter errors resulted in better kinematic parameter error identification and higher pose accuracy.

The effects of pose selection on kinematic calibration are shown in Figures 6–8. Notice that identification of kinematic parameter error obtained by using pose set 1 are consistently worse than those obtained by using pose set 2, even though pose set 1 has more poses than pose set 2. This shows that choosing the poses is more important than the number of poses contained in the pose set. However, very little improvement can be obtained once the number of poses exceeds a certain limit. For the rest of the discussions, kinematic calibration using pose set 3 will be compared, since this pose set yielded good identification results with only 32 poses.

The effect of measurement noise on kinematic calibration is shown in Figures 9 and 10. Notice that pose accuracy improves as measurement noise is reduced, and kinematic parameter errors are perfectly identified when measurement noise is close to zero. This shows that measurement noise should be at least an order of magnitude lower than the desired pose accuracy.

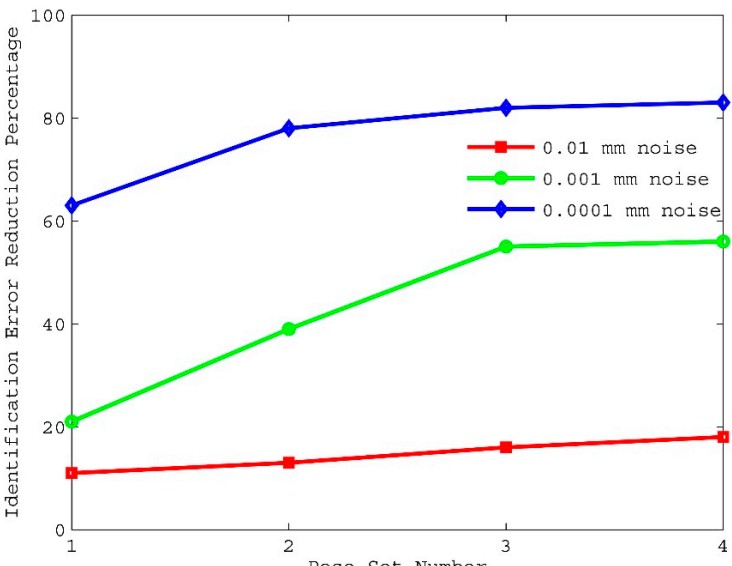

**Figure 6.** Identification error reduction percentage versus pose selection for set I.

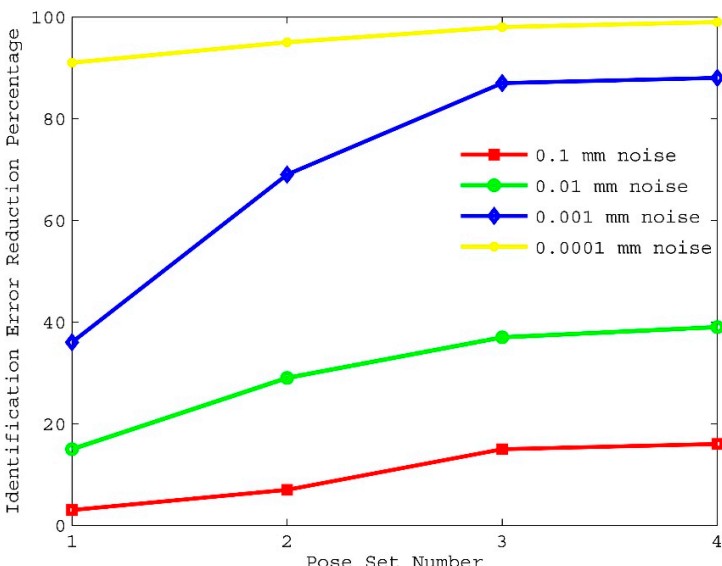

**Figure 7.** Identification error reduction percentage versus pose selection for set II.

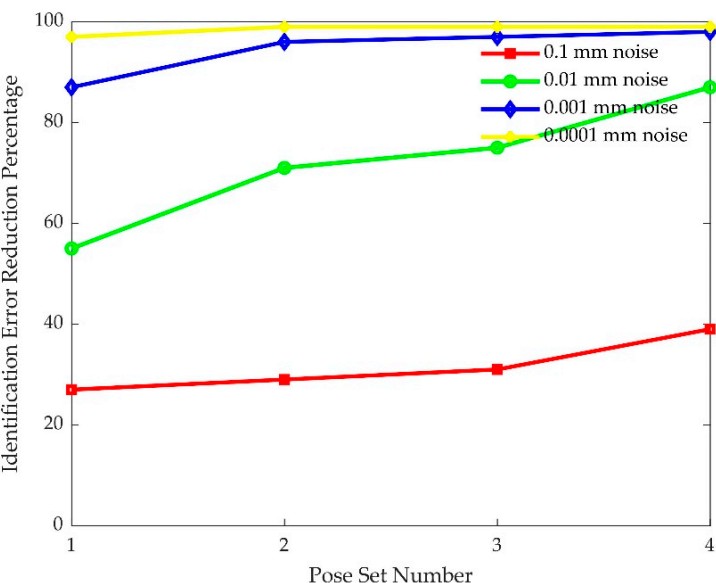

**Figure 8.** Identification error reduction percentage versus pose selection for set III.

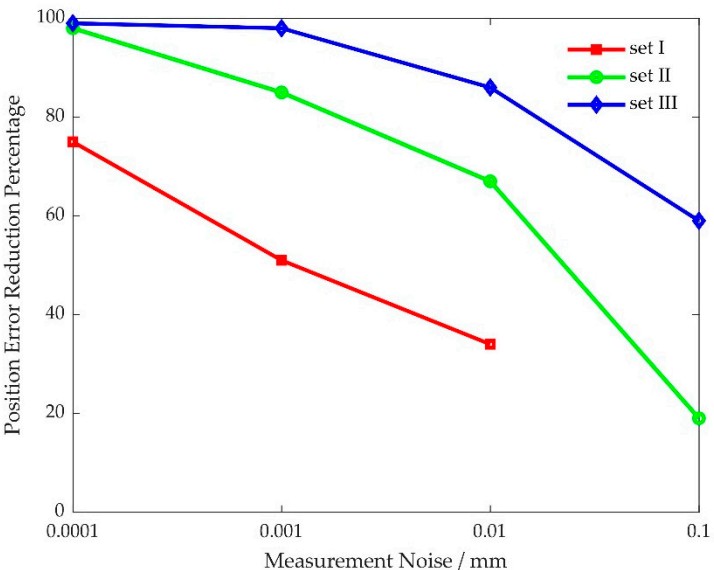

**Figure 9.** Position error reduction percentage versus measurement noise.

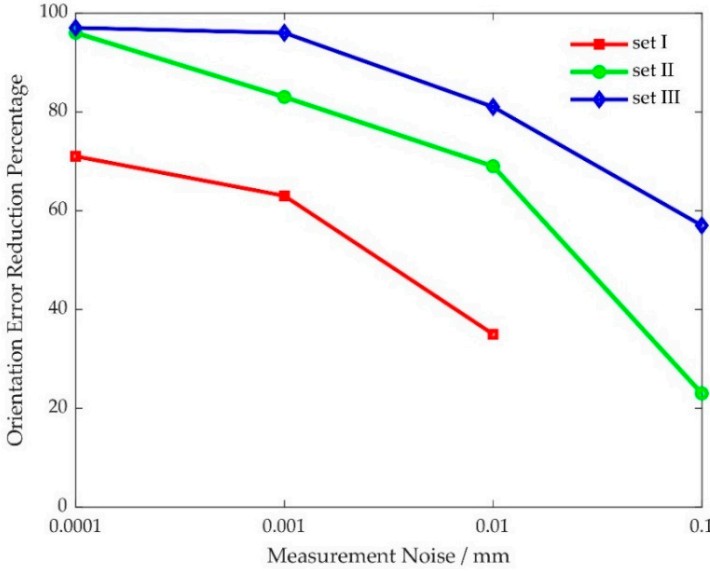

**Figure 10.** Orientation error reduction percentage versus measurement noise.

*5.3. Comparison Experiments*

The experimental system mainly consisted of a parallel robot, a three-dimensional coordinate-measuring machine and six standard spheres. The experimental system for kinematic calibration based on L-infinity parameter estimation is shown in Figure 11. Pose measurement of the parallel robot was done with a precise three-dimensional co-ordinate measuring machine, model 3000i manufactured by STAR Tech. The measuring machine has a point repeatability of 0.010 mm and a length accuracy of 0.016 mm in the 1.2 m × 1.2 m × 1.2 m measuring range. Three of these spheres were fixed at three specific locations of the moving platform, and the other three were fixed at three specific base locations. On this basis, pose measurement of the parallel robot was developed, which mainly measured the distances from three standard spheres on the moving platform to three standard spheres on the base by the coordinate-measuring machine. Then, the poses were computed using these distances, as shown in Figure 12. It is worth mentioning that the kinematic calibration experiments were performed in a limited area due to the length measurement limitation of the coordinate-measuring machine.

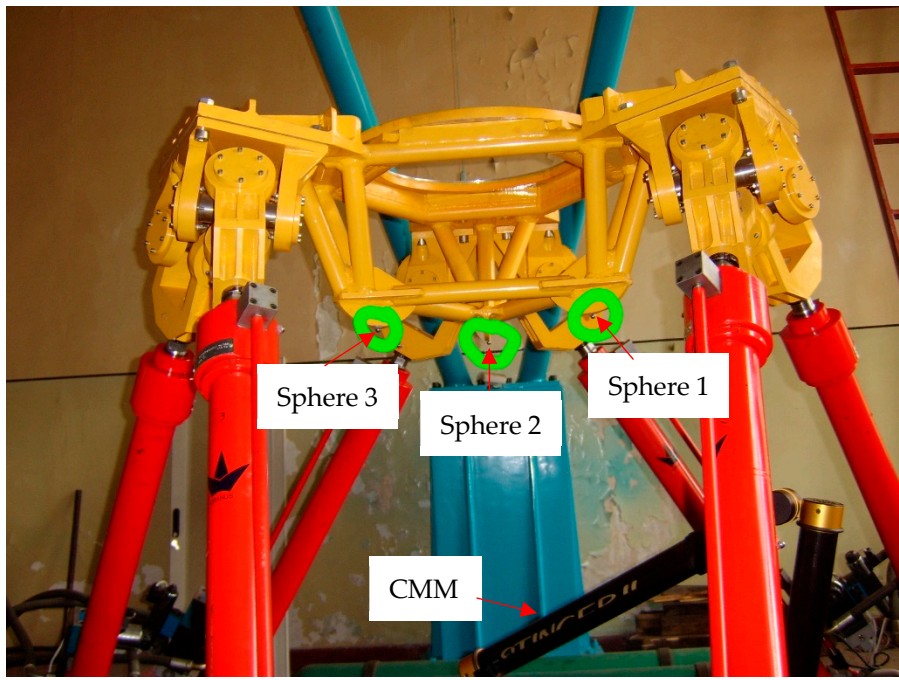

**Figure 11.** The experimental system.

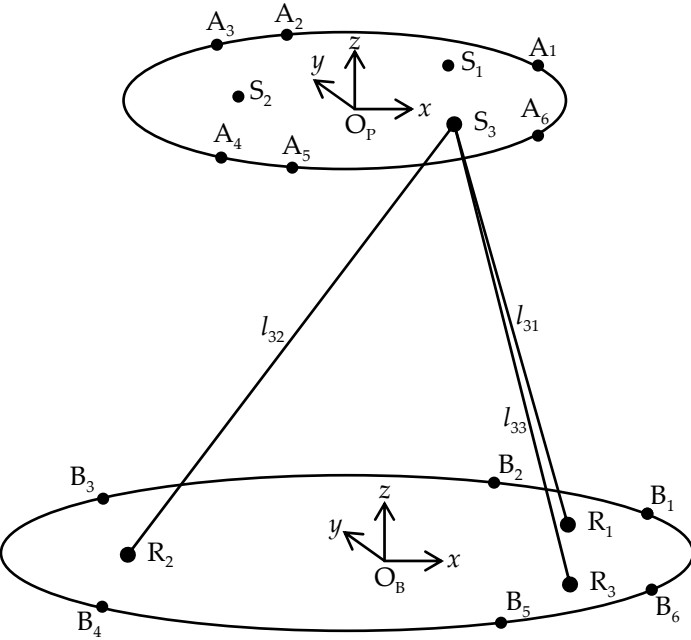

**Figure 12.** The schematic diagram of pose measurement.

The comparison experiments of kinematic calibration were performed using the conventional least squares algorithm and the proposed L-infinity parameter identification algorithm. The two results were compared according to the following four indicators: (1) maximum error, (2) range of error, (3) average error and (4) root mean square error. According to the pose error model, full pose measurement is needed to solve kinematic parameter errors in the two kinematic calibrations. The full pose could be obtained by using a mobile, flexible triad coordinate measuring machine. On the foundation of the abovementioned identification simulations, a measured pose selection rule was determined to make the comparison more effective and to better carryout the experiment: 32 measurement poses based on pose set 3 were chosen to cover the whole workspace by using the

union of a full factorial exploration and a coordinate exchange algorithm in the comparison experiments. Meanwhile, 24 verification poses evenly distributed in the workspace of the parallel robot were also collected to validate pose accuracy improvement in the experiment.

Pose error measurement and kinematic parameter error identification were accomplished by using the original 32 pose errors at the measurement poses listed in Table 6. Then a new round of pose measurement was performed to evaluate pose accuracy after the two kinematic calibrations. The pose errors at verification poses based on the two kinematic calibrations were obtained and are shown in Figures 13 and 14, and the four pose error indicators before and after kinematic calibration are listed in Table 7. The maximum position error after kinematic calibration by using conventional least squares algorithm was 1.0980 mm, and the maximum position error after kinematic calibration by using the proposed L-infinity parameter identification algorithm was 0.9408 mm. The maximum orientation errors were 0.1037 and 0.0848 degree, respectively. The maximum position error based on least squares was reduced by 84.22%, and the maximum position error based on L-infinity was reduced by 86.48%. The maximum orientation errors were reduced by 85.14% and 87.85%, respectively. The range of position error based on least squares is reduced by 85.76%, and the range of position errors based on L-infinity was reduced by 87.01%. The range of orientation errors were reduced by 87.67% and 88.99%, respectively. The percentage reductions of average error and root-mean-square error for a conventional least squares algorithm were almost identical at 83.37% and 83.42%, respectively. The percentage reductions of average error and root-mean-square error for the L-infinity parameter identification algorithm was similar to the conventional least squares algorithm. The two errors were 85.29% and 85.34%, respectively. Clearly, both kinematic calibrations can improve pose accuracy, while kinematic calibration based on the L-infinity parameter identification algorithm is much better than the conventional least squares algorithm. This verifies that the proposed kinematic calibration based on the L-infinity parameter identification algorithm is effective and can achieve strict bounds on the pose errors produced by the parallel robot.

**Table 6.** The pose errors at measurement poses before kinematic calibration.

| | $\delta x$ (mm) | $\delta y$ (mm) | $\delta z$ (mm) | $\delta \omega_x$ (deg) | $\delta \omega_y$ (deg) | $\delta \omega_z$ (deg) |
|---|---|---|---|---|---|---|
| 1 | 0.7286 | −6.3683 | 3.9254 | −0.6993 | 0.0787 | −0.5509 |
| 2 | 3.2461 | 3.7276 | −4.6233 | 0.5205 | 0.1881 | −0.4997 |
| 3 | 0.4247 | −3.5500 | 5.2487 | 0.1637 | −0.6549 | −0.4653 |
| 4 | 7.1205 | −0.7135 | 7.0661 | 0.6958 | 0.1667 | 0.1756 |
| 5 | −3.8926 | 2.7736 | 0.3100 | 0.0440 | −0.1890 | 0.1089 |
| 6 | −5.4966 | −1.8954 | 5.5656 | −0.0239 | −0.6302 | −0.6266 |
| 7 | 5.4412 | 3.4634 | 1.3559 | 0.4299 | −0.0097 | 0.6130 |
| 8 | −6.0964 | −1.3912 | −4.8010 | −0.3787 | −0.4286 | 0.3274 |
| 9 | −1.2509 | 2.7113 | −4.1599 | 0.0023 | 0.5265 | 0.3404 |
| 10 | −0.6286 | 3.0045 | −1.2172 | 0.5702 | −0.4103 | −0.6106 |
| 11 | −1.8018 | −0.7148 | 3.6391 | 0.1103 | −0.4934 | 0.5132 |
| 12 | 3.8494 | −6.7218 | 4.7315 | 0.4917 | −0.4334 | 0.6175 |
| 13 | 1.9224 | −2.2985 | 4.2254 | 0.3415 | −0.6399 | 0.6880 |
| 14 | 3.9698 | −0.9706 | −2.4738 | 0.1262 | 0.1956 | 0.51111 |

**Table 6.** *Cont.*

|    | $\delta x$ (mm) | $\delta y$ (mm) | $\delta z$ (mm) | $\delta\omega_x$ (deg) | $\delta\omega_y$ (deg) | $\delta\omega_z$ (deg) |
|----|----------|----------|----------|----------|----------|----------|
| 15 | 6.2558 | −3.1595 | 0.5891 | −0.3521 | −0.3026 | 0.4076 |
| 16 | 6.8226 | −4.1999 | −5.7218 | 0.2396 | 0.0594 | 0.0239 |
| 17 | −4.2713 | 4.6767 | −5.4127 | −0.5823 | 0.2802 | −0.4496 |
| 18 | −5.0266 | −0.8908 | −5.0633 | 0.1826 | 0.0038 | −0.1380 |
| 19 | 2.8939 | 5.6152 | 2.6436 | 0.2319 | 0.0555 | −0.5112 |
| 20 | −5.6668 | −1.4413 | 0.0365 | 0.3290 | −0.0723 | −0.6564 |
| 21 | 0.4660 | 3.9291 | −4.3042 | 0.5560 | −0.5253 | 0.6242 |
| 22 | 0.5362 | −1.3616 | 0.0340 | 0.6850 | −0.0086 | −0.2752 |
| 23 | 5.2368 | 4.4890 | −4.9025 | 0.3843 | 0.5027 | −0.2833 |
| 24 | −0.1102 | 3.7296 | −6.2188 | 0.1198 | 0.5322 | −0.2306 |
| 25 | −1.4090 | −1.6372 | 5.0886 | 0.6089 | −0.3189 | −0.0414 |
| 26 | 2.5410 | −3.9304 | 0.9656 | 0.1179 | −0.4061 | 0.2140 |
| 27 | 3.5333 | 4.2317 | 6.2097 | −0.6761 | 0.0966 | −0.6644 |
| 28 | 0.3899 | 6.4896 | 2.8996 | −0.5296 | 0.2028 | 0.4875 |
| 29 | −2.0590 | −2.3453 | 1.2815 | 0.5164 | −0.1120 | 0.0882 |
| 30 | −4.8685 | 2.5387 | 4.5868 | −0.0171 | −0.4096 | 0.5043 |
| 31 | 1.3284 | −0.7669 | 5.4908 | 0.4912 | 0.6366 | −0.2095 |
| 32 | −3.2749 | 4.8440 | 7.0524 | −0.4047 | −0.5843 | −0.0711 |

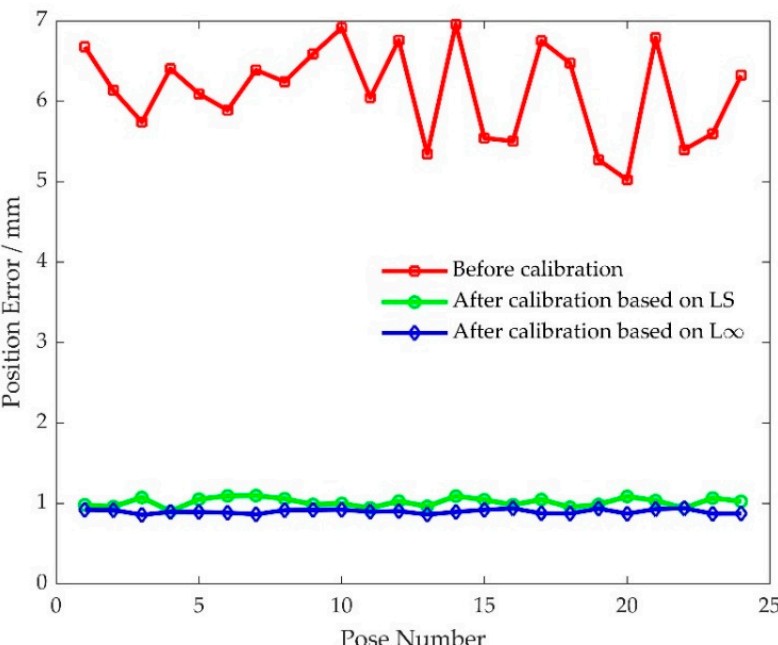

**Figure 13.** The absolute position errors at verification poses before and after calibration.

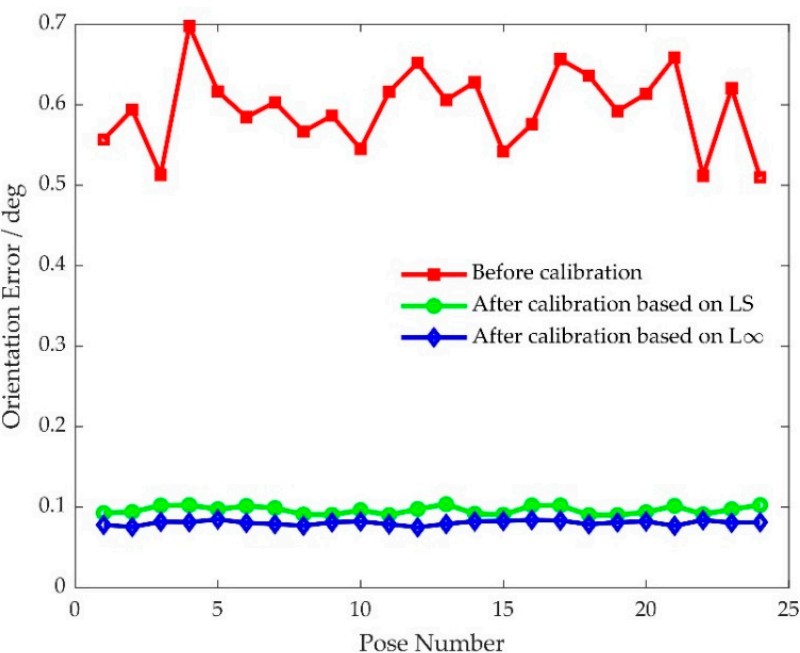

**Figure 14.** The absolute orientation errors at verification poses before and after calibration.

**Table 7.** Comparison of absolute pose errors at verification poses before and after calibration.

| | Before Kinematic Calibration | | After Kinematic Calibration | | | |
| --- | --- | --- | --- | --- | --- | --- |
| | | | Based on L-Infinity | | Based on Least Squares | |
| | Position (mm) | Orientation (deg) | Position (mm) | Orientation (deg) | Position (mm) | Orientation (deg) |
| Maximum error | 6.9595 | 0.6977 | 0.9408 | 0.0848 | 1.0980 | 0.1037 |
| Range of error | 14.2670 | 1.4721 | 1.8534 | 0.1620 | 2.0313 | 0.1815 |
| Average error | 6.1194 | 0.5952 | 0.9001 | 0.0805 | 1.0174 | 0.0964 |
| Root-mean-square error | 6.1455 | 0.5971 | 0.9005 | 0.0806 | 1.0189 | 0.0965 |

## 6. Conclusions

A new kinematic parameter error identification algorithm for the kinematic calibration of a parallel robot using L-infinity parameter estimation is developed in this paper. The kinematic parameter error identification procedure is transformed into a linear programming problem that computes kinematic parameter errors for a pose error model of a parallel robot so that the maximum difference between the predictions and measurements across its workspace is minimized. A strict bound on the pose errors produced by the parallel robot is given in the kinematic calibration based on L-infinity parameter estimation. The experimental results show a 14.32% reduction in maximum position errors and a 18.23% reduction in maximum orientation errors by using L-infinity parameter estimation compared to least squares estimation. The comparison results show an 8.76% reduction in range of position errors and a 10.74% reduction in range of orientation errors by using L-infinity parameter estimation compared to least squares estimation. Therefore, this validates that the proposed kinematic calibration method can effectively improve pose accuracy of the parallel robot and determine the range of the pose error. It should be noted that this kinematic calibration method can be used when pose measurement errors are tightly restricted and measurement noise is low.

**Funding:** This research received no external funding.

**Data Availability Statement:** Not applicable.

**Conflicts of Interest:** The author declares no conflict of interest.

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
