# Peer review of "Kinematic Calibration of Parallel Robots Based on L-Infinity Parameter Estimation"

_machines, doi:10.3390/machines10060436_

Round 1

Reviewer 1 Report

The paper deals with a parameter estimation criterion for kinematic calibration of parallel kinematics robots. In particular, the novelty is the use of the L-infinity norm of the difference between the measured and the computed pose error, in order to identify the kinematic parameter errors.

Some comments on the paper:

  • in the abstract (row 11), and also in other parts of the paper, you mention a "linear program"; maybe you mean the Matlab program that you have created using the LINPROG function, but it isn't clear. I suggest to explain better.
  • At row 152, it would be better to explain the meaning of RPY (i.e., Roll Pitch and Yaw).
  • At rows 157-159, the nomenclature doesn't fit the one in figure 2: in the text the points of the mobile platform and of the base are indicated, respectively, with capital letters A and B, while in the figure you use small letters.
  • In figure 2, I suggest to draw the vector loop represented by equation (3).
  • In my opinion, at row 170 it isn't appropriate to say that hydraulic cylinder control is responsible for the kinematic parameter errors. This is just a problem of the control system performances, not a kinematic configuration issue. It would be a kinematic configuration issue if the length of piston in known positions, like the end stroke ones, would be wrongly evaluated.
  • In section 5 and it isn't so clear that you are talking about just simulations. For example, at row 283, you say "Actuate the parallel robot..", and "...actual driving inputs...". I think that these expressions may generate confusion in the reader who could think that you are referring to the real robot. Moreover, I suggest to clearly say that the kinematic parameter errors of table 2 are just supposed and not real values.
  • I suggest to improve the quality the graphs in figure 3 and 4; they are hardly readable mainly due to the presence of the lines that, moreover, aren't useful.
  • Figure 5, 6 and 7 should be moved after their first citation in the text, that is after row 343.
  • At row 358, please correct "Thee" with "Three".
  • At row 405, I think that "by" should be added before "the parallel robot".
  • At row 416 you say ".....the maximum pose error in its whole workspace is minimized.", but it isn't clear from the results which is the value of the pose error for poses other than the one used for calibration.
  • Finally, I suggest to give a revise to the written text.

Reviewer 2 Report

The article deals with the kinematic calibration of parallel robots based on the criterion of L-infinity for parameter estimation. A novel method for identifying kinematic parameter errors using a linear program is developed, and the resulting pose error model minimizes the maximum pose error of the parallel robot in its workspace to achieve the specified accuracy requirement in each pose throughout the workspace. The article is interesting, but lacks some important details and clear structure, making it difficult to distinguish between the methods used and the results obtained. The experiments are not explained in enough detail, making it difficult for the reader to fully understand the data prediction procedure.

  1. How exactly are the vector t and the matrix R defined? I assume they represent the transformation from OB -xyz to OP -xyz, is that correct?
  2. is it possible to add the vectors t, Ai, Bi, ui, li to figure 2?
  3. in figure 2, Ai and ai are interchanged. My understanding is that they are the same parameter, but once in upper case and once in lower case.
  4. line 197: ui' is undefined?

5: Equation 26 should be written as an equation.

min y =

  1. Line 272: At least 30 measurement poses are required in kinematic calibration of the parallel robot. Please explain why only 30 measured possess were selected and which possess were selected to evaluate 60 independent parameters.

How were the parameters identified?

  1. Line 286: 4. Compute the actual pose errors, namely, subtract the selected poses from the actual poses.

When (in which step) and how do you obtain the actual poses?

  1. Line 288: 5. Compute the pose errors by using the pose error model with the nominal kinematic parameters, the nominal driving inputs, and the kinematic parameter errors.

What are the nominal driving inputs?

  1. Table 2. The kinematic parameter errors.

How are kinematic parameter errors determined?

  1. Line 297: 2. The pose errors computed by using the pose error model are basically consistent with the actual pose errors.

What are actual pose errors? How are they determined?

  1. Line 365: in a limited area

What was the limited measurement range?

  1. The experimental setup is explained without the support of figures, which makes it hard to imagine the actual situation during the experiments. Line 358: Three of these spheres are fixed at three specific locations of the moving platform, and the rest three are fixed at three specific locations of the base

If possible, include figures showing the robot with the spheres attached and the measurement setup on the CMM device.

  1. Line 309: Three kinematic parameter error sets were given, with the kinematic parameter errors obtained from normal distributions with variances of 0.01 mm (set I), 0.1 mm (set II), and 1 mm (set III).

Please explain the following explanations in more detail. If possible, attach the error data as supplementary material in the form of a table.

  1. Line 314: Pose set 1 contains 32 random poses. Pose set 2 contains 24 poses based on a full factorial exploration of the six pose variable limits. Pose set 3 contains 32 poses selected from the workspace using a coordinate exchange algorithm for the optimal experimental design. Pose set 4 contains 64 poses selected by using a coordinate exchange algorithm. Line 379: Meanwhile, 24 poses evenly distributed in the workspace of the parallel robot are also collected to validate pose accuracy improvement in the experiment. Line 382: The pose error measurement and kinematic parameter error identification are accomplished by using the original 32 pose errors

Which poses were chosen for parameter error identification and which for validation? What exactly are the variables and how exactly is the experiment set up. Please provide more details in the following explanations. If possible, attach the data as supplementary material in the form of a table.

Round 2

Reviewer 2 Report

Thank you for your responses. In the revised version, important details have been added and the structure of the article has been improved. The experiments are now well explained. The article is easy to understand, and the reader can understand the methods used and the procedure of data prediction.